# Taxonomic and Metagenomic Survey of a Peat-Based Straw Degrading Biofertilizer

**DOI:** 10.3390/microorganisms13122830

**Published:** 2025-12-12

**Authors:** Grigory V. Gladkov, Anastasiia K. Kimeklis, Olga V. Orlova, Tatiana O. Lisina, Tatiana S. Aksenova, Arina A. Kichko, Alexander G. Pinaev, Evgeny E. Andronov

**Affiliations:** 1All-Russia Research Institute for Agricultural Microbiology, Saint Petersburg 196608, Russia; grgladkov@arriam.ru (G.V.G.); akimeklis@arriam.ru (A.K.K.);; 2Zoological Institute of Russian Academy of Sciences, Saint Petersburg 199034, Russia; 3Department of Applied Ecology, Saint-Petersburg State University, Saint Petersburg 199034, Russia

**Keywords:** biofertilizer, plant residue, decomposition, amplicon sequencing, metagenome sequencing, nitrogen metabolism, sulfur metabolism, straw, peat

## Abstract

The mobilization of complex microbial communities from natural resources can be a valuable alternative to the use of single-species biofertilizers when it comes to the decomposition of plant residues. However, the functioning and interaction of microorganisms within these communities remain largely unexplored. Our task was to investigate the cellulose-degrading community using the biofertilizer BAGS (peat-based compost with straw) as an example and define its active component. For this, we monitored the succession of the biofertilizer’s taxonomic composition during two consecutive rounds of its six-month composting process, varying in the applied mineral fertilization. The amount of added nitrogen significantly affected the performance of the biofertilizer, contributing to its high cellulolytic activity. Based on the network analysis, the biofertilizer’s mature phase was determined, and its characteristic ASVs (amplicon sequence variants) were described. Metagenomic analysis of this phase revealed MAGs (metagenome-assembled genomes) corresponding to these ASVs, which contained genes for cellulose and aromatics degradation, as well as genes for nitrogen and sulfur pathways, including anaerobic nitrate reduction and thiosulfate oxidation. Thus, we propose that the cellulose-decomposing bacterial component of BAGS, associated with the mature phase, occupied different trophic niches, not limited to cellulose degradation, which should be considered when designing natural or artificial microbial systems for the decomposition of plant residues.

## 1. Introduction

Lignocellulose degradation remains a major ecological and agricultural problem, especially with current crop production rates [1,2,3]. In 2025, worldwide plant residue generation is estimated to reach two billion tons [4]. Most often, this biomass is viewed as waste and burned, which leads to the emission of pollutants like carbon dioxide, sulfur dioxide, and particulate matter [5]. Agricultural waste is therefore a huge source of carbon, which should be sequestered in soil organic matter rather than returning it to the atmosphere as carbon dioxide and increasing the greenhouse effect. Adding straw to arable land is a more ecological way of its utilization, which improves soil organic matter and structure [6,7,8]. The increasing soil organic carbon, in turn, may enhance nitrogen turnover and reduce its runoff [9]. Straw return can be accomplished by various approaches, including mulching and shredding [10]; however, direct addition of straw can serve as an inoculum of plant pathogenic microbiota [11]. This problem can be avoided by using biofertilizers in concordance with straw application. Besides, carbon storage in organic compounds can be more efficient using biofertilizers capable of assimilating highly recalcitrant cellulose fibrils.

Plant residues mainly contain long-chain carbohydrates, such as cellulose and hemicellulose. They also contain noncarbohydrates, like lignin and pectin [12]. The degradation of cellulose fibrils is accomplished by three types of enzymes: (1) endoglucanases, which break the internal bonds in cellulose chains; (2) exoglucanases, which remove two or four glucose units from the chain’s free end; and (3) glucosidases (cellobiases), which produce single glucose units [13,14]. The degradation of hemicellulose, lignin, and other compounds requires even more enzymes. These enzymes, catalogued in the CAZy (Carbohydrate-Active enZYmes) database, form over 100 families categorized by class, structure, and substrate specificity [15]. Apart from cellulose decomposition, composting involves biochemical changes to elements essential to microbes, such as nitrogen, phosphorus, and sulfur [16,17,18].

Due to the complexity and variability of plant residues as substrates, their digestion requires multienzyme complexes that are distributed among numerous members of a decomposing microbial consortium [19,20]. Nevertheless, many studies focus on isolating individual cellulase-producing strains [21,22,23,24,25]. Most of these strains belong to a narrow list of taxa; for example, the most common fungi in industry are *Trichoderma* spp. and *Aspergillus* spp. [26,27,28,29,30], and the most common bacteria—*Bacillus*, *Pseudomonas*, *Streptomyces*, and *Arthrobacter* [31,32,33].

While fungi are the main players in cellulose decomposition in natural ecosystems, they are more challenging to cultivate and produce on an industrial scale [34]. On the contrary, bacteria offer functional diversity, flexible demand for oxygen, and tolerance to a wide range of environmental conditions [35]. Thus, although bacterial biofertilizers may be less effective than fungal ones, they may have an advantage in terms of simplicity and cost of development [36].

This study investigates the microbial composition of the complex biofertilizer BAGS (the abbreviation is originally Cyrillic and has no direct translation). The story of creating BAGS dates back more than a century, when the concept of autochthonous soil microbiota was introduced by Vinogradsky [37,38]. Using this concept, the first iteration of the microbial consortium, called AMB (“autochthonous microflora B”), was developed in the 1930s, and was produced by composting peat, mineral fertilizer, and humate-decomposing microbial inoculum isolated from soil [39,40]. By the 1990s, the concept and method of composting had shifted, and a new biofertilizer, BAGS, was developed. Like AMB, it was based on peat composted with mineral fertilizer, but with the addition of straw and cellulolytic microbial inoculum. Originally, one of the goals of this biofertilizer was to degrade pesticides, but eventually this function was omitted. For over 30 years, this biofertilizer was preserved at the ARRIAM (All-Russia Research Institute for Agricultural Microbiology) through a series of subcultures and, before now, no one had addressed the question of the composition and functional properties of its microbiota. Considerable work has however been carried out to determine its influence on the decomposition process, with a strong effect of BAGS on the microbiome of straw compost being reported [38,41].

It is not yet known whether BAGS is a permanent microbial consortium, which is preserved through a series of subcultures, or whether it should be considered merely a technology for selecting cellulolytic microbiota from natural resources. Each round of BAGS preparation introduces fresh unsterilized peat from the ombrotrophic bog and cellulosic material (primarily oat straw) to the mix, which requires at least six months of maturation before use. Earlier this year, using taxonomic data from one generation of BAGS, it was shown that its microbiome was unaffected by the microbiomes of the peat and straw, and that the composition of 11-month-matured compost resembled that of the inoculum [38]. The BAGS microbiome described in that study comprised over a thousand taxa, so even with its overall stability, it is not surprising that with such species richness, its composition may be subject to change depending on local composting conditions. This is how the present study was devised, where we compared two consecutive generations of BAGS prepared with minor technological differences, namely with differing amounts of added nitrogen.

The aims of the study were to study the taxonomic succession of the microbial consortium during BAGS composting, and describe the metabolic potential of its mature phase. To achieve this, we measured the cellulolytic activity of two BAGS generations, performed sequencing of 16S rRNA gene amplicon libraries from different phases of BAGS incubation, and performed full metagenomic sequencing of its maturation phase. The hypothesis was that cellulolytic activity and changes in microbial composition would help us identify the active phase of the biofertilizer and its associated microbiota. Metagenomic analysis, in turn, would reveal the diversity of the functional profile of this potentially active microbiota. This knowledge would deepen the understanding of interaction within complex microbial cellulose-degrading communities.

## 2. Materials and Methods

### 2.1. The Experiment Layout

In the current study, we sampled two consecutive batches of BAGS, made in 2023–2024 (Appendix A). The first batch (BAGS2023, gen1) was prepared in May 2023, using the inoculum from 2022, and the second batch (BAGS2024, gen2) was prepared in November 2023, using the product from the first batch as the inoculum. The quantity of components used in each batch is described in Table 1. Absolute values of components fluctuated between batches due to differences in moisture content in the peat. The main difference between the batches was the amount of added nitrogen: in the first batch we added ammonium nitrate taking into account the amount of nitrates introduced with the peat, while in the second batch we disregarded the nitrates in the peat and added all nitrogen, phosphorous and potassium (NPK) salts, weighing 1% of the straw’s dry weight. The amount of nitrogen introduced was adjusted for gen2, taking into account the effectiveness of the gen1 biofertilizer.

Before combining the components, the straw was soaked in 1.3 L of tap water with the solution of NPK salts. After that, all components were combined in a 90 L plastic tub and composted in a thermostatically controlled room at a temperature of 28 °C for six months. The composting product was periodically stirred and watered to maintain moisture content. Each BAGS batch was represented by one container, from which we performed an equivalent sampling scheme: starting from the first day, a single sample of about 100 g every 30 days for six months, in total 7 samples for each generation. Part of the sampled material was used immidiately for cellulolytic activity analysis, while the rest was frozen for subsequent amplicon and metagenomic analysis.

### 2.2. Testing the Cellulolytic Activity of BAGS

To compare the efficiency of cellulose degradation by microbial consortia from two BAGS generations, we assessed the cellulolytic activity of the BAGS compost via decomposition of a paper filter. For this, a thin layer of BAGS compost was spread in a Petri dish, covered with filter paper, watered, and incubated at 28 °C for one month. The maceration of the paper filter was estimated visually. We tested actual and potential cellulolytic activity. For the latter, the BAGS was additionally stimulated with mineral nitrogen.

### 2.3. The DNA Isolation and Sequencing

For the amplicon sequencing, we used seven monthly samples from each BAGS batch, from day 0 to 180. From each sample, total DNA was extracted in quadruplicate using the RIAM protocol [42], resulting in 56 DNA samples. The DNA for the metagenome analysis was extracted using the same method from three-, four-, and five-month-old samples from the second BAGS batch (gen2). These samples were selected based on the performance of this batch, as tested by microbial and taxonomic analyses. Preparation and sequencing of amplicon libraries of the 16S rRNA gene on the Illumina MiSeq (Illumina, Inc., San Diego, CA, USA) and full metagenome on the MinION (Oxford Nanopore Technologies, Oxford, UK) were performed as described earlier [43]. On Illumina, we performed 2 × 300 bp pair-end sequencing of amplicon libraries, obtained with primers F515 (5′-GTGCCAGCMGCCGCGGTAA-3′) and R806 (5′-GGACTACVSGGGTATCTAAT-3′) [44,45]. The full metagenome library was sequenced on the R10.4.1 flow cells (FLO-MIN114) using the Native Barcoding Kit 24 V14 (SQK-NBD114.24) (Oxford Nanopore Technologies, Oxford, UK).

### 2.4. The Amplicon Data Analysis

The 16S rRNA gene amplicon sequencing data were processed in R v. 4.3.2 [46] using the dada2 v. 1.28.0 pipeline [47] as described earlier [43,48]. The alpha diversity within rarefied by the smallest amplicon library samples was estimated as the number of amplicon sequence variants (ASVs), and beta-diversity between unrarefied samples was visualized by non-metric multidimensional scaling (NMDS) [49] with Bray-Curtis distances [50]. ANCOM-BC v. 2.2.2 (analysis of compositions of microbiomes with bias correction) was performed to detect characteristic taxa between BAGS batches [51]. The WGCNA v. 1.73 (weighted gene coexpression network analysis) [52] was applied to identify groups of covarying ASVs throughout the maturation of the BAGS microbial community. A co-occurrence network of ASVs within both BAGS generations was constructed based on the SPIEC-EASI v. 1.1.3 algorithm [53]. The post-processing of the resulting graph was performed as described earlier [54].

### 2.5. The Metagenome Data Analysis

The metagenome analysis was performed using a genome-centric approach. The base calling of the metagenome sequencing data was performed using the Dorado neural network model v. 5.00 in super accuracy mode [55]. The obtained sequences were assembled using the Flye program in the --meta mode [56]. The read alignment for the assembly pipeline was performed using the minimap2 v. 2.28 program [57]. The binning was performed using the Semibin2 v. 2.0.2 program [58]. High- and medium-quality MAGs (Metagenome Assembled Genomes) were selected from the general pool using the CheckM2 v. 1.0.2 program [59]. Bins were filtered according to the following rule: completeness should be higher than 50%, contamination less than 10%, and the difference Completeness—Contamination*5 should be higher than 50, which corresponds to the quality standards of the GTDB database. Taxonomic annotation of MAGs was carried out using the gtdb-tk v. 2.4.0 software pipeline [60]. Functional annotation was carried out using the eggnog-mapper program with flags --diamond and --very-sensitive [61] and METABOLIC v. 4.0 [62]. A search of functional categories associated with CAZy genes was performed using modified DRAM annotation [63] (Appendix A), and a search of PULs (polysaccharide utilization loci) was done using dbCAN v. 4.1.4 [64]. The results were visualized using R (tidyverse v. 2.0.0 [65], ggplot2 v. 3.5.2 [66]) and Python v. 3.10.13 (polars v. 0.20.31) software (Polars BV, Maarssen, The Netherlands). The data are deposited under the NCBI Project ID PRJNA1345249.

## 3. Results

### 3.1. The Comparison of the Cellulolytic Activity of the Two BAGS Generations

When testing the efficiency of BAGS for the ability to decompose filter paper, gen1 only developed high cellulolytic activity after five months of composting, unlike gen2, which already showed cellulolytic activity in the first month (Appendix A). At the same time, when measuring potential activity (with the introduction of additional ammonium nitrogen), gen1 had cellulolytic activity in the fourth month. This data coincides with the difference in nitrification activity between the two BAGS generations.

### 3.2. The Comparison of the Taxonomic Composition of the Two BAGS Generations

#### 3.2.1. The Diversity of the Two BAGS Generations

The amplicon sequencing included 56 samples, which resulted in 673,519 reads. The analysis confirmed differences between the BAGS generations not only at the level of chemical composition, but also at the microbiome composition level. Firstly, alpha diversity of the gen1 BAGS rose during the six months of composting, while in gen2, it dropped (Figure 1). This effect was evident both in the richness (observed ASV count) as well as in the evenness of the community (Inverted Simpson index). The beta diversity shows similar trends for both generations of BAGS: the microbiome shifted along the *x*-axis, and this shift became less significant over time. For the gen1 BAGS, this shift persisted throughout the entire six months of composting, while for gen2, starting from the fourth month, the community stabilized. Notably, there was no continuity of the microbial community across BAGS generations: the chronoseries were significantly distant from each other (R^2^ = 0.304, *p*-value < 0.001).

#### 3.2.2. The Major Bacterial Components of Two BAGS Generations

Taxonomic analysis of both BAGS generations revealed common trends in the shifts of the microbiome at the bacterial phylum level during composting (Figure 2). Generally, the relative abundance of Pseudomonadota, Actinomycetota, and Bacillota dropped, while Chloroflexota, Patescibacteria, and Myxococcota rose. Both generations were abundant in Bacteroidota and Acidobacteriota. Among the pronounced differences between generations, it is worth noting the increased Bacillota and Actinomycetota content in the first months of composting the ineffective gen1 BAGS, while for the effective gen2 BAGS, a high representation of Bacteroidota and Patescibacteria phyla was more characteristic of the later stages of composting.

#### 3.2.3. The Differences in Bacterial Composition Between BAGS Generations

Taxonomic markers of the two BAGS generations were assessed by the ANCOM-BC (Appendix A). The analysis revealed 516 ASVs, of which 291 increased in the first generation and 225 in the second. This shift is most likely due to the higher diversity of the gen1 microbiome. The first, less effective gen1 BAGS with insufficient nitrogen was characterized by the following microorganisms: Bacteroidota (*Flavobacterium hauense*, *Flavitalea*, *Mucilaginibacter*, *Adhaeribacter terreus*, Saprospiraceae, *Cellulomonas*), Pseudomonadota (*Massilia oculi*, *Lysobacter soli*, *Caulobacter*, *Rhizobium*), Actinomycetota (*Arthrobacter alpinus*, *Conexibacter*, *Galbitalea*), Nanoarchaeota (AR15), Eremiobacteriota, Acidobacteriota (Subgroup 10), WS2, Patescibacteria (Candidatus *Nomurabacteria*, *Parcubacteria*). In the second, effective gen2 BAGS, the following groups of microorganisms significantly (*p*-adj < 0.05, lfc > 1) increased their representation: Patescibacteria (Gracilibacteria), Chloroflexota (*Anaerolinea*), Pseudomonadota (MND1, *Amphiplicatus*, *Tahibacter aquaticus*, *Burkholderia oxyphila*, *Burkholderia silvatlantica*, *Nordella*, *Rhizobium phaseoli*), Verrucomicrobiota (*Oikopleura*, *Pedosphaera*), Bacteroidota (*Chryseolinea*, *Terrimonas*, *Edaphobaculum*, *Chryseotalea*), Thermodesulfobacteriota, Latescibacterota, Chlamidiota (Simkaniaceae), Myxococcota (Haliangiaceae), Hydrogenedentes (Hydrogenedensaceae), Planctomycetota (Pirellulaceae, *Planctomicrobium*), Acidobacteriota (*Luteitalea*, *Paludibaculum*).

#### 3.2.4. The Network Analysis of BAGS Microbiome

The most pronounced differences in taxonomic composition between gen1 and gen2 were observed in the late stages of the maturation process, as shown by the network analysis (Appendix A). Based on the WGCNA, we allocated clusters of covarying ASVs those abundance was rising during the composting process (Appendix A, brown cluster). Both clusters contained *Terrimonas*, IS-44 (Seq119 and Seq421 from Pseudomonadota), and Seq447—unidentified Myxococcota from family Haliangiaceae (Appendix A). In gen1, they emerged only after five months of composting, while in gen2 they became relatively abundant after three months of composting. The most prominent distinctive feature of the cluster from the effective gen2 BAGS was the presence of a pair of Seq636 (Chryseolinea, Bacteroidota) and Seq106 (an unidentified Gracilibacteria, Patescibacteria). Other ASVs present belong mostly to phyla Pseudomonadota (Seq919, Seq91, Seq1069, Seq614), Planctomycetota (Seq1825), Chloroflexota (Seq83), Hydrogenedentota (Seq140, Seq1483), and Acidobacteriota (Seq804, Seq1484). Several ASVs belong to rare microbiota from Methylomirabilota, Verrucomicrobiota, Thermodesulfobacteriota, Dependentiae, and Bdellovibrionota.

The co-representation of Gracilibacteria and Chryseolinea in the chronoseries of the biofertilizer was confirmed using co-occurrence networks (Figure 3). They were constructed both for each generation of BAGS separately (Appendix A) and for the two generations together (Figure 3A). Topologically, the networks of gen1 and gen2 BAGS were similar to each other. The modularity value for gen1 BAGS was 0.4, for gen2 BAGS—0.46; the clustering coefficient was 0.065 and 0.093, and the edge density was 0.015 and 0.018, respectively. The network with gen2 was slightly more heterogeneous, with more pronounced modules than with gen1. In the combined network for two BAGS generations, a subgraph of microorganisms correlating with Gracilibacteria (Seq 106) was identified (Figure 3A). Of the 11 bacteria in this subgraph, Chryseolinea was the most represented (Figure 3B).

Thus, the combined results of microbiological and taxonomic analyses of two batches of the BAGS biofertilizer showed that BAGS effectiveness depends on the amount of nitrogen introduced. A set of microorganisms associated with the BAGS mature phase, was described, among which a pair of Chryseolinea and an unidentified Patescibacteria of the Gracilibacteria family was identified. Therefore, for further functional analysis of the mature BAGS, samples of gen2 after four months of composting were selected for metagenome sequencing.

### 3.3. The Metagenomic Analysis of the Mature Phase of Gen2 BAGS Biofertilizer

#### 3.3.1. The Assembly of the Metagenome

The extraction of nucleic acids from BAGS was challenging due to the high content of peat humic compounds. Of the four runs of sequencing using ONT technology, only two metagenomes of acceptable quality from 4- and 5-month composting samples were obtained. After base calling, 32.24 Gb were obtained with a median quality of 12.82 and N50 = 37,438. The length of the longest read was 1.52 Mb. The final assembly of the two metagenomes was 1.78 Gb with N50 = 34,271.

From this assembly, 81 medium-quality metagenome-assembled genomes (MAGs) with an average coverage of 124 were extracted. The MAGs included representatives of the phyla Pseudomonadota (29 bins), Acidobacteriota (16 bins), Planctomycetota (6 bins), Chloroflexota (six bins), Myxococcota (four bins), Gemmatimonadota (four bins), Patescibacteriota (three bins), Bacteroidota (two bins), Desulfobacterota (two bins), Hydrogenedentota (two bins), Methylomirabilota (two bins). Phyla Actinomycetota, Bdellovibrionota, Chlamidiota, Eisenbacteria, and Verrucomicrobiota were represented by a single MAG each. Most MAGs aligned with ASVs obtained from amplicon data (Appendix A). A heatmap of these ASVs confirms that most of them represent microorganisms present in the mature phases of the generation of the effective gen2 BAGS (Appendix A, Appendix A). Among the most numerous microorganisms for which MAGs have been obtained were Gracilibacteria (bin_22, Seq106), *Chryseolinea* (bin_42, Seq636), Anaerolineae (bin_25, Seq22), Chitinophagaceae (bin_4, Seq119), Burkholderiaceae (bin_1, Seq2), Burkholderiales (bin_0, Seq919), Tepidisphaeraceae (bin_33, Seq1825), Steroidobacteraceae (bin_26, Seq29), and Dongiaceae (bin6, Seq48). Below, we analyze the possible functional roles of isolated MAGs in BAGS, based on their annotation. Among the most promising systems, we considered genes for carbohydrate, nitrogen, and sulfur metabolism, as well as genes for antifungal activity and the breakdown of aromatic compounds.

#### 3.3.2. Long-Chain Carbohydrate Metabolism

As genomic markers of cellulolytic activity, we considered both the representation of individual families of CAZy genes (according to DRAM annotation) in MAGs and their combination into polysaccharide utilization loci (PULs) (as predicted by db-CAN) (Figure 4). Only those PULs that were specialized on substrates with β-1,4 linkages between sugar residues were included in the analysis, as they are most likely associated with the degradation of complex polysaccharides. Based on the high presence of individual CAZy genes, MAGs from the following taxa were distinguished: bin_21 (Polyangiaceae), bin_33 (Tepidisphaeraceae), bin_38 (Vicinamibacteria), bin_12 (*Polyangia*), bin_15 (*Polyangia*), bin_25 (Anaerolineae), bin_50 (Vicinamibacteria), bin_141 (*Peristeroidobacter*), bin_4 (Chitinophagaceae), and bin_42 (*Chryseolinea*).

Aside from the total number of CAZy genes, a likely indicator of the ability to degrade a high diversity of cellulosic substrates is the number of PULs, specifically relative to the genome size [67,68,69]. In our dataset, two MAGs with such properties were present: bin_26 (Steroidobacteraceae) and bin_33 (Tepidisphaeraceae) (Figure 5). Both MAGs contained 13 predicted PULs and were relatively small—4.8 Mb and 5.2 Mb, respectively.

#### 3.3.3. Aromatics Metabolism

In addition to the functional annotation of genes associated with the degradation of complex carbohydrates, other metabolic pathways associated with the decomposition of plant residues were also considered. The genes of laccases, polyphenoloxidases (AA1, AA2), were identified as marker genes of ligninase activity. These genes were identified in 50 bins and were most common in bin_21 (Polyangiaceae) and bin_9 (Gemmatimonadales) (Figure 5, Appendix A). Genes for the degradation of various cyclic compounds, including xenobiotics (pathways for the degradation of aromatics, polycyclic compounds, furfural, dioxin, benzonate), were identified in several MAGs, such as representatives of Xanthobacteraceae (bin_14) and Cupriavidus (bin_138).

#### 3.3.4. Nitrogen Metabolism

We detected genes of nitrogen metabolism in 55 out of 81 MAGs (Figure 5, Appendix A). Nitrification genes were only detected in two MAGs. Ammonium oxidation (K10535, hao) was detected in bin_48 (Desulfobacterota) and nitrite oxidation (K00370, K00371, *nxr*A, *nxr*B) in bin_138 (*Cupriavidus*). On the contrary, genes of nitrate/nitrite reduction were plentiful. The gene of nitrous oxide reductase (K00376, *nosZ*), used as a marker of the denitrification pathway, was present in 13 MAGs (Appendix A). The nitrate reduction pathway is used not only in denitrification, but also in the DNRA (dissimilatory nitrate reduction to ammonium). We used the presence of nitrite reductase (*nirB*, K00362) as a marker of fermentative DNRA. Fermentative DNRA pathway genes were identified in 24 MAGs. For respiratory DNRA, we analyzed the abundance of nitrite reductase (K03385, *nrf*A and K15876, *nrf*H). These genes were present in 12 MAGs. Most of these MAGs were predicted to be key microorganisms of the effective phase of BAGS. Among those were bin_33 (Tepidisphaeraceae) and bin_21 (Polyangiaceae), which were also rich in CAZy genes.

#### 3.3.5. Sulfur Metabolism

Sulfur metabolism genes were found among the components of the BAGS biofertilizer. In the community we studied, thiosulfate to sulfite oxidation genes were found in six MAGs: bin_35 (Micropepsaceae), bin_34 (Planctomycetota), bin_57 (Vicinamibacteria), bin_16 (Rokubacteriales), bin_25 (Anaerolineae) and bin_5 (Haliangiaceae) (Figure 5, Appendix A). Genes of the SOX complex (thiosulfate to sulfate oxidation) were present in 25 MAGs and were more common in Burkholderiales (bin_1, bin_79, bin_108, bin_0), Casimicrobiaceae (bin_2), and Xanthobacteraceae (bin_14).

To sum up, among 81 MAGs, most of which are characteristic of the mature phase of BAGS, we observed metabolically diverse groups of microorganisms. While some MAGs specialized solely in the decomposition of cellulose chains and their derivatives, like bin_4 or bin_27, in others, genes associated with straw decomposition were found to coexist with genes for aromatics, nitrogen, or sulfur metabolism, like bin_26, bin_33, bin_42, bin_17, bin_25, and bin_21. There was also a group of non-cellulolytic MAGs in which we didn’t detect an overrepresentation of CAZy, but which had auxiliary functions, like bin_1, bin_138, or bin_35. Notably, there was a group of MAGs that corresponded to ASVs designated as BAGS mature phase markers, but no target functional group genes were found in them. Among those was bin_22, belonging to Gracilibacteria, which aligned with one of the major ASVs in the cluster associated with decomposition.

From the composition and analysis of the functional potential of the MAGs found in the mature active phase of the BAGS biofertilizer, we can draw several conclusions. First, the BAGS microbial community is a metabolically heterogeneous group, the various functions of which are distributed among different taxa. Second, of the entire community, only a small proportion (representatives of Steroidobacteraceae, Tepidisphaeraceae) were predicted to be solely cellulolytic microorganisms, and the rest had mixed metabolic potential or belonged to the associated microbiota. The functional specialization was rather distributed across different MAGs. Some of the associated microbiota carried genes for the nitrogen and sulfur cycles that were not present in the cellulolytic fraction.

## 4. Discussion

The study focused on explaining how microbiota interact in the formation of a cellulose-degrading community, using the example of the BAGS biofertilizer. To achieve this, we tested the conditions for composting the biofertilizer and determined the composition and function of the bacteria associated with the mature phase of the biofertilizer.

### 4.1. The Conditions for the Effective gen2 BAGS

Each round of BAGS biofertilizer is essentially a reactivation of cellulolytic microbiota from the previous generation by adding a fresh source of cellulosic material, thus triggering the new decomposition process. But the introduction of plant residues, specifically straw, creates an excess of carbon relative to nitrogen [70]. The optimal decomposition C:N ratio lies between 20:1 and 30:1, which ensures an equilibrium between mineralization and immobilization processes [71]. To achieve this ratio, mineral nitrogen fertilizer is usually added in the amount of 1% of the dry weight of the incorporated straw [72]. Consistent with this, previous studies have shown the cumulative effect of microbial treatment and nitrogen fertilization on straw decomposition rates [36]. The first generation of BAGS, described here as gen1, was prepared with the recommended N treatment but also with consideration of the nitrates contained in the introduced peat. As a result, it showed low cellulolytic activity. Most probably, the deficiency in nitrogen led to its rapid immobilization in microbiota [10]. In terms of microbial community, nitrogen deficiency facilitated the increase of microbial alpha-diversity, which contradicts earlier findings [73]. Presumably, the nitrogen deficit facilitated the growth of a diverse oligotrophic microbiota, none of which was highly specialized in cellulose degradation. Thus, the microbiome of the first generation of BAGS was used as a control to help us identify the active component of the effective generation of BAGS from 2024 (gen2).

After improving the methodology by adding higher quantities of nitrogen, we obtained a biofertilizer whose cellulolytic activity was most noticeable after four months of composting. This coincides with the succession of ASV clusters in the amplicon data: the microbial composition of BAGS stabilized after four months of composting, and at later stages it changed insignificantly. Similar succession in the microbiome from straw-based compost had been reported previously [43,48,74,75].

Amplicon and metagenomic analyses showed convergence of results and made it possible to isolate the microbiome representatives most associated with the mature phase of the effective BAGS biofertilizer and propose metabolic processes in which they were most likely involved.

### 4.2. Functional Diversity of the BAGS Microbial Community

#### 4.2.1. The Cellulolytic Microbiota

The bacterial composition of the mature phase of BAGS included more than 700 ASVs. Despite this, among the MAGs, we isolated only two microorganisms with an overrepresentation of cellulolytic clusters that break down complex polysaccharides—a representative of the Steroidobacteraceae (Pseudomonadota) and a representative of the Tepidisphaeraceae (Planctomycetota). Overrepresentation of PULs is not a prerequisite for an active cellulolytic but may nevertheless indicate such [68]. Steroidobacteraceae are not described as classical cellulolytic microorganisms, but there is evidence that they possess cellulolytic potential [76]. Representatives of Tepidisphaeraceae are also described in the literature as being able to break down complex polysaccharides [77]. This group is poorly represented in the literature due to the fact that it mostly refers to uncultured microorganisms [78].

#### 4.2.2. The Accompanying Microbiota

As an example of the microbiota associated with true cellulolytic microorganisms, we would like to highlight the MAG from Myxococcota of the family Haliangiaceae. It has a very large genome (13.81 Mb) and a high representation of many metabolic pathways, including carbohydrate decomposition. These microorganisms are described as predatory bacteria with a complex metabolism [79,80]. Here, as well as in our previous studies of decomposing microbial communities, we have noted a high representation in the community of specific predatory microbiota, such as Bdellovibrionota [43,48]. The role of predatory microbiota in such communities has not been studied, but its importance in community stabilization [81] and the selection of specific microbiota is obvious.

Some of the accompanying microbiota possess rare metabolic pathways that are not present in the cellulolytic part of the community. Their role in the community is often unclear to us, but since this microbiota remains present from one generation of BAGS to the next, we can assume the role of these microorganisms in stabilizing the community.

#### 4.2.3. The Nitrogen and Sulfur Metabolism

Although our experiment wasn’t directly designed to test the effect of nitrogen on the effectiveness of decomposition, it was shown to be an important factor in determining the cellulolytic efficiency of BAGS as a bioproduct. Indeed, the gen2 BAGS, which was prepared with added nitrogen, had a significantly higher potential nitrification level than the gen1 BAGS. This suggests that BAGS activity is likely to be directly related not only to the cellulolytic component of the community, but to microorganisms associated with the nitrogen cycle. The nitrogen availability was shown to be linked with the cellulase activity during litter decay [70]. Therefore, we analyzed the presence of key nitrogen metabolism genes associated with nitrogen retention in MAGs [82,83].

For communities with a high C:N ratio, typical of cellulolytic communities, the dissimilatory nitrate reduction to ammonium (DNRA) pathways play an important role [83] as they may predominate over denitrification [84]. It can be assumed that the difference in the two BAGS generations is associated with a shift in the balance between denitrification and DNRA in favor of the latter, which leads to nitrogen conservation in the effective BAGS.

Wang and Li demonstrated that, in composts, enzymatic DNRA and the consequent abundance of the *nir*B and *nir*D genes inversely correlate with the abundance of respiratory DNRA, depending on the decomposition stage [85]. At the same time, enzymatic DNRA is more characteristic of thermophilic conditions and the early stages of decomposition. In our experiment, decomposition occurred at room temperature, and enzymatic DNRA pathways were more abundant in the assembled genomes than respiratory DNRA pathways. Furthermore, the abundance of DNRA pathways was higher than that of key denitrification enzymes (K00376, *nos*Z), further indirectly indicating the importance of DNRA for the effectiveness of the gen2 BAGS.

The balance between denitrification and DNRA can also be influenced by other factors, such as the presence of sulfides [86]. This is supported by the presence of sulfide:quinone oxidoreductase (sqr, K17218), characteristic of both respiratory DNRA and sulfite oxidation [87], in a minor microorganism from bin_16 of the effective gen2 BAGS (a representative of the phylum Methylomirabilota). The presence of sulfate reducers in the community may be a marker of the importance of sulfur metabolism in cellulolytic communities, as indicated earlier.

#### 4.2.4. Non-Specialized Microbiota

One of the key markers of the effective BAGS was the presence of Gracilibacteria from the Patescibacteria superphylum, representatives of which reached up to 13% of the total relative abundance of microbiota in mature biofertilizer. According to published data, representatives of the superphylum Patescibacteria are capable of forming ectosymbiosis with bacteria from Bacteroidota [88]. As we noted earlier, it was this phylum that showed differences in representation depending on the phase of decomposition. Moreover, its representation within a single generation correlated with the representation of Chryseolinea. In the analyzed MAGs, the representative of Gracilibacteria was characterized by a small genome of 1.4 Mb (the genome size of the putative Chryseolinea host was 7.3 Mb). It lacked CAZy genes, cytochromes, and chemotaxis genes. Compared with the genome of the putative host, it contained genes of the type II secretory system, indicating the possibility of attachment to the host or the secretion of exoproteins [89]. Also, among the genes present in the Gracilibacteria genome but not present in the Chryseolinea genome were: acetate metabolism pathways (acetogenesis, *acd*A/ack/pta), F-type ATPase (*atp*AD), peptidases M50B, S16, genes of pyrimidine synthesis pathways, genes of iron reduction pathways, and sulfur metabolism genes (*lux*S, *mtn*N). The Chryseolinea genome was not oversaturated with PUL but contained two putative xylan cleavage clusters and one chitin cleavage cluster. In comparison with other studied genomes, it contained a significant number of glycoside hydrolase (GH) families associated with the cleavage of beta-galactan (GH2, GH35, GH42, GH1, GH59, GH147, GH165, GH53). The possibility of acetate synthesis by Gracilibacteria can be associated with the possibility of its influence on the ratio of nitrification and DNRA [90], which could significantly affect the balance of nitrogen forms in the community [91]. Based on the data obtained, we cannot predict the role of these microorganisms in the community, but their overrepresentation was characteristic specifically of gen2 BAGS, which was more effective than the previous one.

### 4.3. The Anaerobic Properties of BAGS

Several observations indicate that BAGS composting created conditions for the development of microaerophilic or even anaerobic microbiota. The first predisposition to this effect was the C:N ratio. While the C:N range for the effective composting remains quite wide, the higher end of the C:N range (around 30:1) promotes aerobic decomposition, while the lower (below 20:1)—anaerobic [92,93]. At the beginning gen2 BAGS had a C:N range of 23:1 (Appendix A), but by the end, only 17:1. This would also explain the extensive presence of genes for the denitrification, because it was shown that under anaerobic conditions, denitrification processes may prevail over nitrification [94]. Presumably, it is due to the microaerophilic conditions in BAGS that we discovered the DNRA pathway in nitrogen metabolism. Lastly, the decomposition of straw in anaerobic environments, which include composts, often occurs with the cooperation of several microorganisms, some of which are cellulolytic, and others are utilizers of cellulose decomposition products [95]. Among such microbiota, it is worth noting the presence of MAGs of poorly studied representatives of the phyla Hydrogenedentota and Desulfobacterota [96,97]. The presence of these microorganisms may also indicate the anaerobic nature of the metabolism of part of the microbiome. Supposedly, the incubation process created conditions for the partial anoxia, specifically the stratification of the peat-straw compost into the aerobic top and anoxic deep layers.

## 5. Conclusions

The mobilization of cellulolytic microbiome from natural resources is a promising technique for creating biofertilizers, since it can select bacteria with broad metabolic functions. BAGS is an example of such a complex microbial consortium with a specific microbiota—originally isolated from soil, but since diverged from it. There is a stable core microbiota that reproduces between generations of BAGS, but its composition remains flexible, depending on the composting conditions. Here we have shown that the amount of nitrogen introduced is one of such conditions, which intensified the cellulolytic activity of BAGS.

The combination of cellulolytic activity and network analysis allowed us to determine the mature phase of the BAGS, which formed after the fourth month of composting. We identified the active bacterial content of the mature phase of effective BAGS, which, according to the MAGs’ analysis, included both potentially purely lignocellulolytic (e.g., Steroidobacteraceae, Tepidisphaeraceae, with high amount of PULs, or Polyangiaceae with high amount of AA family genes) and accompanying microbiota associated with nitrogen and sulfur cycles. In addition, microorganisms unrelated to any of these functions were associated with the active phase of the biofertilizer (e.g., a potential cellulolytic Chryseolinea with its presumable symbiont Gracilibacteria with an unknown function).

Using the example of BAGS, we demonstrated that in complex biofertilizers, the microorganisms associated with the lignocellulose degradation may have different functions, the roles of which remain to be determined. Perhaps, knowledge of the synergy of these functions will aid in the design of highly effective biofertilizers.

## Figures and Tables

**Figure 1 microorganisms-13-02830-f001:**
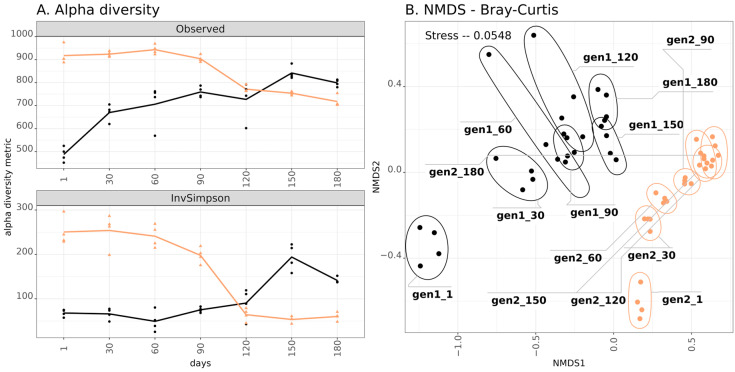
(**A**) Alpha diversity of two BAGS generations expressed in Observed index (number of ASVs) and Inverted Simpson index. Lines express mean values: black for gen1 BAGS (2023), orange for gen2 BAGS (2024). Replicate measurements (4 for each sample) are shown as black circles for gen1 and orange triangles for gen2 BAGS. The *x*-axis on the left denotes days of incubation. (**B**) Beta diversity of two BAGS generations visualized on the NMDS plot with Bray-Curtis distances. Amplicon libraries of replicate samples are encircled. The inscription includes generation number and days of incubation.

**Figure 2 microorganisms-13-02830-f002:**
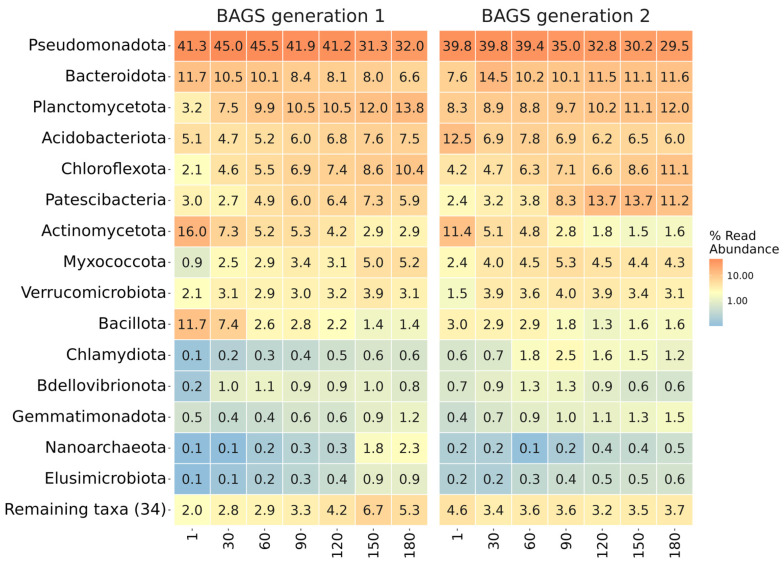
The taxonomic composition of BAGS generations on the phylum level. Heatmap shows relative abundance of ASV within each phase. Orange color denotes higher values; blue—lower. The numbers under columns denote days of incubation.

**Figure 3 microorganisms-13-02830-f003:**
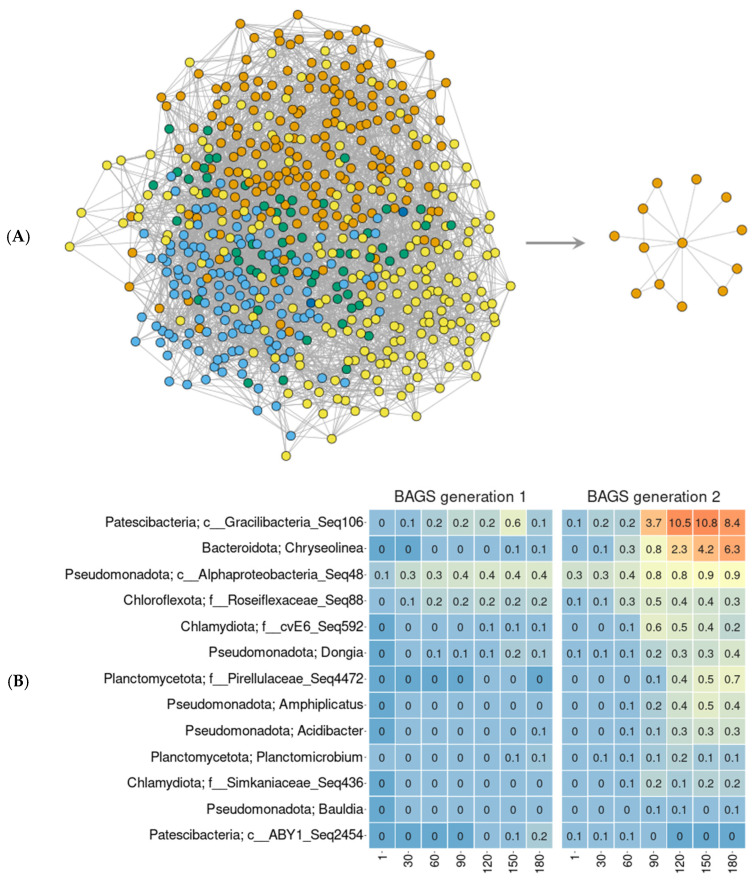
The network analysis (SPIEC-EASI algorithm) of combined BAGS generations (**A**, **left**) and a subgraph of ASVs, connected with Gracilibacteria (**A**, **right**), and their relative abundance in each generation (**B**). The differently colored dots on the graphs denote different co-representation ASV modules. Heatmap shows relative abundance of ASV within each phase. Orange color denotes higher values; blue—lower. The numbers under columns denote days of incubation.

**Figure 4 microorganisms-13-02830-f004:**
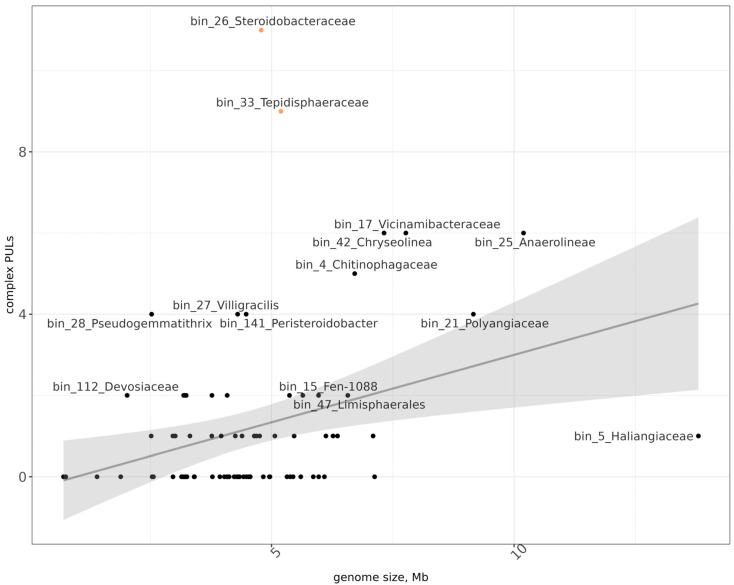
Distribution of MAGs (black dots) according to the number of complex PULs and their genome sizes. x-axis—genome size in Mb, y-axis—number of PULs. Orange dots mark two MAGs with relatively high to the genome size number of PULs.

**Figure 5 microorganisms-13-02830-f005:**
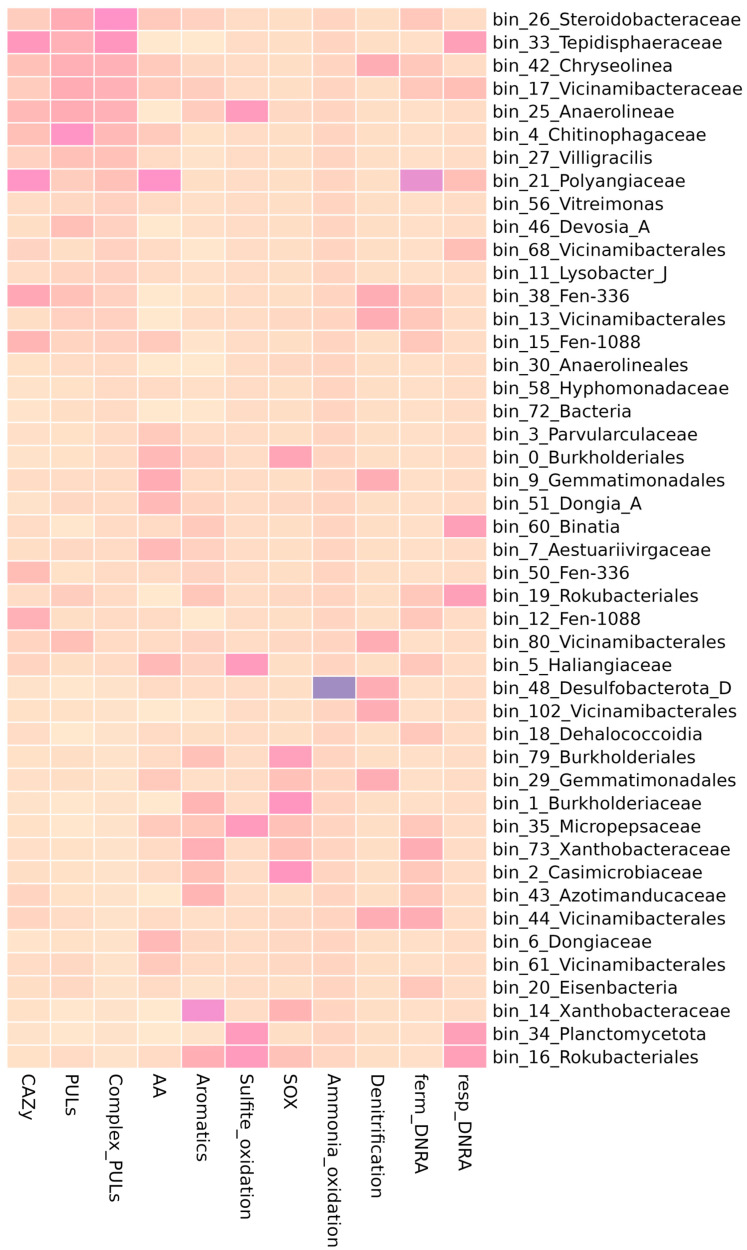
The functional annotation of MAGs corresponding with the cluster of ASVs from the mature phase of BAGS. Heatmap shows the relative abundance of genes from selected metabolic pathways within each column. CAZy—genes for carbohydrate-active enzymes, PULs—polysaccharide utilization loci, Complex PULs—PULs with the substrate specificity towards lignocellulose components, AA—genes from auxiliary activity family of CAZy, Aromatics—genes of aromatics decomposition, SOX—genes for thiosulfate to sulfate oxidation, DNRA—genes for dissimilatory nitrate reduction to ammonium.

**Table 1 microorganisms-13-02830-t001:** The components of BAGS biofertilizer in two rounds of composting.

Component	BAGS2023 (gen1)	BAGS2024 (gen2)	Description
Boreal peat (wet)	15 kg	13.3 kg	Equal to 4 kg dry weight
BAGS inoculum (wet)	3.34 kg	3.09 kg	20% of the wet substrate weight
Dolomite	0.28 kg	0.37 kg	7% of the peat’s dry weight
Oat straw (air-dried)	0.42 kg	0.42 g	Equal to 400 g dry weight, which is 10% of the peat’s dry weight
NH_4_NO_3_	4.8 g	13.7 g	Equal to 4 g of N, which is 1% of the straw’s dry weight
K_2_HPO_4_	2.4 g	3.3 g	1% of the straw’s dry weight
KH_2_PO_4_	4.2 g	5.8 g	1% of the straw’s dry weight

## Data Availability

The data presented in this study are openly available in NCBI BioProject ID number PRJNA1345249.

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
