# Peer review of "Taxonomic and Metagenomic Survey of a Peat-Based Straw Degrading Biofertilizer"

_microorganisms, 2025, doi:10.3390/microorganisms13122830_

Round 1
Reviewer 1 Report
Comments and Suggestions for Authors
The authors presented research on shifts in microbial community during composting. The combination of chemical analysis and genomics allowed for enhanced the results. Nitrogen and Sulfur Metabolism were correlated with Microbial shifts. The data would be more impressive if total nitrogen and total sulfur were provided. Although the manuscript as a whole makes a favorable impression, there are some comments that, if addressed, would improve the presentation of the results.
Line 29. What do cyclic compounds mean in the context of the study?
Line 109 and 111. Taxonomic succession and functional orientation are unclear and aren’t mentioned further in the manuscript.
Line 132. Was it because of gen 2 inoculate was enriched with high cellulitic activity microbes or due to the changes in mineral composition only?
Line 492. What kind of water was applied?
Line 504. What was a method of ash measuring (dry ash, sulfuric ash?)
Author Response
The authors presented research on shifts in microbial community during composting. The combination of chemical analysis and genomics allowed for enhanced the results. Nitrogen and Sulfur Metabolism were correlated with Microbial shifts. The data would be more impressive if total nitrogen and total sulfur were provided. Although the manuscript as a whole makes a favorable impression, there are some comments that, if addressed, would improve the presentation of the resuls.
We thank the reviewer for the overall good evaluation of our manuscript. We agree that data on sulfur and nitrogen content would improve the interpretation of metagenomic data. However, as it usually happens, chemical analysis was performed before the metagenome analysis, so we didn’t foresee the necessity of sulfur quantification. As for the nitrogen, we have data, but only for the second batch of preparation. We mentioned this data (line 466), but only briefly, since we couldn’t apply it to the whole dataset.
Line 29. What do cyclic compounds mean in the context of the study?
In our study we focused on the metabolic pathways of plant residues decomposition, and we assume that the major component of plant residues is lignocellulose. But we try to address the fact that lignocellulose is a complex compound, consisting not only of cellulose, but hemicellulose and lignin. While genes for cellulose decomposition, coding glycoside hydrolases, are mostly united in CAZy, the ligninases are not so compactly organised. While ones can be found in the CAZy database in the AA family (e.g. AA1, AA2), there are still other genes of enzymes, connected with lignin decomposition, but not included in the CAZy. Some of them are enlisted in the Supplementary information (Table S4). Because these enzymes are targeted at the cyclic part of the molecule, we generalised this section as metabolism of “cyclic compounds”. For better clarity we renamed it as “Aromatics”.
Line 109 and 111. Taxonomic succession and functional orientation are unclear and aren’t mentioned further in the manuscript.
The aims and scope have been reformulated.
Line 132. Was it because of gen 2 inoculate was enriched with high cellulitic activity microbes or due to the changes in mineral composition only?
The question is valid. Since gen2 used BAGS from gen1 as the inoculate, it is fair to assume that it is changes in mineral composition that drove this activity. But we can’t be 100% sure since we compare 2 batches of the preparations, which weren’t designed to be parts of one experiment.
Line 492. What kind of water was applied?
unsterilised tap water. The information was added to the manuscript.
Line 504. What was a method of ash measuring (dry ash, sulfuric ash?)
It was dry ash method (direct combustion at 800 degrees in a muffle furnace for 2 hours). The information was added to the manuscript.
Reviewer 2 Report
Comments and Suggestions for Authors
Dear Authors,
Overall, I think this manuscript is very interesting. It's also well-prepared scientifically. However, I have a few corrections you should consider, which will improve the quality of this article.
Please clarify your comments below:
Line 45: Do you mean burning plant-based straw—this is how carbon dioxide is released into the atmosphere? It's also worth mentioning that plant-based straw is shredded (mainly by the company during grain harvesting). It also serves as a mulching material and even as the main raw material for producing mushroom substrates. Only then should you move on to the information that, despite this, even more straw should be added to the soil.
Line 111: You could also specify what the utilitarian goal was in this case. This makes sense, as the research seems very practical in the long term.
Figure 1 and Figure 2: I assume the results were obtained from an average. I suggest adding error whiskers to the graph. I must admit that the variability is quite large, but this will better illustrate the process.
Figure 6: This figure is difficult to read (mainly the descriptions on the right side). It might be worth posting the figures one below the other.
Line 479: Does the materials and methods section have to be at the end of the article? I must admit that this is very unusual. Consider moving this section higher in the article.
Table 1: What causes the unequal mass values, e.g., 3.34 kg, 0.28 kg, 0.37 kg, etc.? You should explain this.
Line 559: Please review the article point by point and check that you have included all the conclusions from your research. Also add one forward-looking, more forward-looking conclusion.
Author Response
Dear Authors,
Overall, I think this manuscript is very interesting. It's also well-prepared scientifically. However, I have a few corrections you should consider, which will improve the quality of this article.
We thank the reviewer for the high evaluation of our manuscript and useful commentary.
Please clarify your comments below:
Line 45: Do you mean burning plant-based straw—this is how carbon dioxide is released into the atmosphere? It's also worth mentioning that plant-based straw is shredded (mainly by the company during grain harvesting). It also serves as a mulching material and even as the main raw material for producing mushroom substrates. Only then should you move on to the information that, despite this, even more straw should be added to the soil.
Thank you for clearing this part. Indeed, we simplified the description of straw recycling, since it was not a primary topic of our research. Actually, we think that straw shredding and mulching can serve as examples of straw return into arable land. We altered the beginning of the introduction accordingly.
Line 111: You could also specify what the utilitarian goal was in this case. This makes sense, as the research seems very practical in the long term.
It seems that our research is aimed at practical application, but I think it is actually more fundamental-oriented. So we modified the last paragraph of introduction:
“The aims of the study were to define the mature phase of the BAGS preparation, reveal its active bacterial component, and describe its metabolic potential. To achieve this, we measured the performance of two BAGS generations using chemical and microbiological methods, performed sequencing of 16S rRNA gene amplicon libraries from different phases of BAGS incubation, and performed full metagenomic sequencing of the maturation phase of the preparation. The hypothesis was that changes in chemical parameters would help us identify the active phase of the preparation and its associated microflora. Metagenomic analysis, in turn, would reveal the diversity of the functional profile of this potentially active microflora. This knowledge would reveal the understanding of interaction within complex microbial cellulose-degrading communities.”
Figure 1 and Figure 2: I assume the results were obtained from an average. I suggest adding error whiskers to the graph. I must admit that the variability is quite large, but this will better illustrate the process.
Based on our experience, we consciously opt out of adding whiskers to graphs with a small number of replicates. Instead, we add dots representing individual values of replicates. We added them on Figure 1 and they already were on figure 2.
Figure 6: This figure is difficult to read (mainly the descriptions on the right side). It might be worth posting the figures one below the other.
Done. The figure is divided in two.
Line 479: Does the materials and methods section have to be at the end of the article? I must admit that this is very unusual. Consider moving this section higher in the article.
I agree, this is very inconvenient. This was the requirement of the other MDPI journal. The manuscript was transferred to Microorganisms without the possibility to reformat the sections. The methods were moved before the results.
Table 1: What causes the unequal mass values, e.g., 3.34 kg, 0.28 kg, 0.37 kg, etc.? You should explain this.
Absolute values of components fluctuated between batches due to differences in moisture content in the peat. The information was added to the manuscript.
Line 559: Please review the article point by point and check that you have included all the conclusions from your research. Also add one forward-looking, more forward-looking conclusion.
The conclusions have been updated according with the recommendations:
“The mobilization of cellulolytic microbiome from natural resources is a promising technique for creating biopreparations, since it can select bacteria of a broad metabolic function. BAGS is an example of such complex multispecies preparation with a specific microbiota, once isolated from soil, but since diverged from it. Even between generations the microbiome of BAGS remains flexible, depending on the composting conditions. While some bacterial members transferred between generations of BAGS, their abundance shifted according to the changes in the external environment; in the case of this study, it was the amount of nitrogen introduced, which intensified the cellulolytic activity of BAGS.
The fluctuation of chemical parameters, like ash content and respiration values, allowed us to determine the mature phase of the BAGS, which formed after the fourth month of the composting. We identified the active bacterial content of the mature phase of effective BAGS, which included both purely cellulolytic (e.g., Steroidobacteraceae, Tepidisphaeraceae, Polyangiaceae) and accompanying microflora associated with nitrogen and sulfur cycles, lignin degradation, and antifungal activity. In addition, microorganisms unrelated to any of these functions were associated with the active phase of the biopreparation (e.g., a potential cellulolytic Chryseolinea with its presumable symbiont Gracilibacteria with an unknown function).
Using the example of BAGS, we demonstrated that in complex microbial preparations, the microorganisms associated with degradation of lignocellulose may have different functions, the roles of which remain to be determined. Perhaps, knowledge of the synergy of these functions will aid in the design of highly effective microbial preparations.”
Reviewer 3 Report
Comments and Suggestions for Authors
The authors present the manuscript entitled “Search for the Active Bacterial Component of the Complex 2 Biological Preparation BAGS.” However, several considerations must be addressed.
First, the authors mention the acronym BAGS in the title but do not define it there, providing its definition only later in the introduction. The abstract itself does not present a clear explanation of what BAGS refers to. In general, the abstract is coherent, but it contains some issues regarding logical flow and certain statements.
The authors should correct this. The abstract does not offer a structured description that matches the experimental organization later described in the manuscript. For example, the authors state that they “identified its most active phase,” yet they do not clarify what exactly this “most active phase” represents.
They also mention in the abstract that “a pair of microorganisms stood out… one cellulolytic (Chryseolinea) and the other its symbiont (Gracilibacteria).” However, in the main text, there is no clear demonstration of this pairing or the nature of their interaction.
Furthermore, the authors conclude that “the microbiota associated with the mature phase of the effective preparation occupied different trophic niches, not limited to cellulose degradation.” This conclusion appears somewhat superficial.
Regarding the introduction, the authors present a comprehensive and well-referenced background that adequately contextualizes the study. Most references cited are from the last five years. However, some references are more than 30 years old, such as Halsall & Gibson (1985) [15], Halsall & Gibson (1989) [16], Bylinkina (1963) [39], and Shul’gina (1941) [40].
The results are well presented and well discussed.
In the conclusion, when the authors state that “The microbiome of BAGS remains flexible between generations,” they appear to overgeneralize. While the manuscript shows differences between generations, it also identifies key groups that remain stable. The conclusion places more emphasis on change than on the persistence of core components. This point should be revised.
When the authors claim that “Their abundance shifted according to the amount of nitrogen introduced,” they indeed present data showing nitrogen effects, but they do not fully isolate this factor from other environmental variables inherent to composting. The conclusion simplifies a relationship that is more complex.
Finally, the authors list purely cellulolytic families (Steroidobacteraceae, Tepidisphaeraceae, Polyangiaceae) based solely on metagenomic analysis, without providing any experimental validation to support this classification.
Author Response
The authors present the manuscript entitled “Search for the Active Bacterial Component of the Complex 2 Biological Preparation BAGS.” However, several considerations must be addressed.
First, the authors mention the acronym BAGS in the title but do not define it there, providing its definition only later in the introduction. The abstract itself does not present a clear explanation of what BAGS refers to. In general, the abstract is coherent, but it contains some issues regarding logical flow and certain statements.
You are right about abbreviations. We opted to remove the abbreviation of the biopreparation from the title, since it is not central to our conclusions.
The authors should correct this. The abstract does not offer a structured description that matches the experimental organization later described in the manuscript. For example, the authors state that they “identified its most active phase,” yet they do not clarify what exactly this “most active phase” represents.
We agree that this part was somehow ambiguous. Since BAGS preparation harbours a multispecies community, which undergoes taxonomic succession during the process of its maturing, we propose that “active phase” would describe microbiota, most possibly connected with the actual decomposition process. We distinguished this microflora based on the succession of the chemical parameters and the taxonomic composition of BAGS preparation during composting. We resembled this in the abstract and at the end of introduction.
They also mention in the abstract that “a pair of microorganisms stood out… one cellulolytic (Chryseolinea) and the other its symbiont (Gracilibacteria).” However, in the main text, there is no clear demonstration of this pairing or the nature of their interaction.
You are right. The Chryseolinea-Gracilibacteria pair is not a central part of the study, it is merely an interesting finding during our analysis. So we removed it from the abstract.
Furthermore, the authors conclude that “the microbiota associated with the mature phase of the effective preparation occupied different trophic niches, not limited to cellulose degradation.” This conclusion appears somewhat superficial.
We shifted our conclusions a little bit, so they would be more accentuated around the description of the active BAGS component. Our main goal was to select from thousands of inhabitants such bacteria that may be directly linked to cellulose degradation. And we think that it is central to our research that not all of these microorganisms are actually connected with cellulose degradation.
Here is the modified conclusion:
“Thus, we propose that the active bacterial component of BAGS, associated with the mature phase of effective preparation, occupied different trophic niches, not limited to cellulose degradation, which should be considered when designing natural or artificial microbial systems for the decomposition of plant residues.”
Regarding the introduction, the authors present a comprehensive and well-referenced background that adequately contextualizes the study. Most references cited are from the last five years. However, some references are more than 30 years old, such as Halsall & Gibson (1985) [15], Halsall & Gibson (1989) [16], Bylinkina (1963) [39], and Shul’gina (1941) [40].
Some of the references refer to the history of BAGS, so they are indeed old. Other references, referring to the metabolism of the plant residues, were revised to be more current.
The results are well presented and well discussed.
Thank you for the high evaluation of our work.
In the conclusion, when the authors state that “The microbiome of BAGS remains flexible between generations,” they appear to overgeneralize. While the manuscript shows differences between generations, it also identifies key groups that remain stable. The conclusion places more emphasis on change than on the persistence of core components. This point should be revised.
Yes, we understand that part of the microbial community was stable, as we underlined in the sentence “some bacterial members transferred between generations of BAGS”. But it was our intention to address the flexibility of such communities. This part was rewritten to be more concise.
When the authors claim that “Their abundance shifted according to the amount of nitrogen introduced,” they indeed present data showing nitrogen effects, but they do not fully isolate this factor from other environmental variables inherent to composting. The conclusion simplifies a relationship that is more complex.
Truly, in our experiment we can’t strictly divide the nitrogen effect from other fluctuations, which happened between generations. We tried to address this issue:
“Here we have shown that the amount of nitrogen introduced is one of such conditions, which intensified the cellulolytic activity of BAGS.”
Finally, the authors list purely cellulolytic families (Steroidobacteraceae, Tepidisphaeraceae, Polyangiaceae) based solely on metagenomic analysis, without providing any experimental validation to support this classification.
Yes, our analysis is purely hypothetical. We clarified it in the conclusions:
“We identified the active bacterial content of the mature phase of effective BAGS, which, according to the MAGs’ analysis, included both purely cellulolytic (e.g., Steroidobacteraceae, Tepidisphaeraceae, Polyangiaceae) and accompanying microflora associated with nitrogen and sulfur cycles, lignin degradation, and antifungal activity.”
Reviewer 4 Report
Comments and Suggestions for Authors
Abstract is too extended (294 words). Please comply with MDPI guidelines
on line 93-94 authors claim "extensive work" but only one reference. Please improve the writing
at the end of the introduction, the authors claim "The aims of the study were to investigate the patterns of taxonomic succession in the BAGS microbiome during composting, to identify microflora connected with its mature stage, and to describe their functional orientation". BUT the title does not reflect this objective. Please change the document title.
Please improve the quality of Figure 1.
The data has no statistical error. Do the samples have no replicates?
Authors show nitrogen as the driver of effectiveness (pp. 118–138 and Discussion 409–427), but the rationale is poor. Please add at least 2–3 literature references describing how C:N ratio drives stabilization vs. microbial competition.
The amplicon figures rely heavily on visual heatmaps and do not include effect size, CI, or directionality tables.
Authors need to support their claims better with scientific references. Why is ash a proxy for decomposition?. Why does the respiration peak mark maturity?. Why nitrogen availability influences cellulolysis mechanistically?. Please support these claims better.
The authors claim that the Gracilibacteria–Chryseolinea is central to the results. However, there is no direct metabolic coupling analysis, and only presence/absence data are available. Please reformulate this claim.
Figures 2A and 2B require zooming to interpret; please re-scale them.
Figure 3 heatmap needs legend scaling
Figures 4A and AB requires heavy explanation on why they are important to the results.
Author Response
Abstract is too extended (294 words). Please comply with MDPI guidelines
The abstract was shortened according to the Microorganisms requirements.
on line 93-94 authors claim "extensive work" but only one reference. Please improve the writing
Corrected. You are right, we worked a lot, but didn’t publish a lot about it.
at the end of the introduction, the authors claim "The aims of the study were to investigate the patterns of taxonomic succession in the BAGS microbiome during composting, to identify microflora connected with its mature stage, and to describe their functional orientation". BUT the title does not reflect this objective. Please change the document title.
Thank you for pointing out this discrepancy. Actually, we opted not to modify title, but to rewrite the last paragraph of introduction:
“The aims of the study were to define the mature phase of the BAGS preparation, reveal its active bacterial component, and describe its metabolic potential. To achieve this, we measured the performance of two BAGS generations using chemical and microbiological methods, performed sequencing of 16S rRNA gene amplicon libraries from different phases of BAGS incubation, and performed full metagenomic sequencing of the maturation phase of the preparation. The hypothesis was that changes in chemical parameters would help us identify the active phase of the preparation and its associated microflora. Metagenomic analysis, in turn, would reveal the diversity of the functional profile of this potentially active microflora. This knowledge would reveal the understanding of interaction within complex microbial cellulose-degrading communities.”
Please improve the quality of Figure 1.
Corrected.
The data has no statistical error. Do the samples have no replicates?
Repetitions have been added to the graph as dots. Since most of the repetitions were two, or the measurements matched, we do not consider that calculating any statistics here would be correct.
Authors show nitrogen as the driver of effectiveness (pp. 118–138 and Discussion 409–427), but the rationale is poor. Please add at least 2–3 literature references describing how C:N ratio drives stabilization vs. microbial competition.
The effect of nitrogen as a driver of effectiveness was revised in several places of discussion. Mostly it is mentioned in the paragraph 4.1
The amplicon figures rely heavily on visual heatmaps and do not include effect size, CI, or directionality tables.
We have corrected Table S2 in the Supplement, which contains the results of statistical analyses (ANCOMBC). We tried other ways of visualizing the differential analysis, but in our opinion, a heat map is the simplest and most understandable way to represent such data. To prove that the text is correct, we provide a modified summary of the results in the table.
Authors need to support their claims better with scientific references. Why is ash a proxy for decomposition?.
Compost consists of the organic and mineral components. Combustion eliminates organic component and leaves ash, which is a mineral component of the compost. The higher percent of ash content means the lower percent of organics in the composting biomass. So, during decomposition of organic compounds, ash content in compost is rising. The changing in ash content allows to estimate the rate of decomposition process. I don’t know, whether this should be a part of the methods section, but I added a reference to this explanation.
Why does the respiration peak mark maturity?.
We address this issue in the following sentence:
“We propose that this was the most effective phase of the BAGS composting period, since a decrease in the amount of nitrogen, particularly nitrates, as well as in respiration rates, has been shown to serve as an indicator of compost maturity [79,80]. ”
Respiration (or CO2 emission) is an indicator of carbon mineralization rate. So, the higher respiration, the more active are decomposing bacteria. The explanation was added in the materials section.
Why nitrogen availability influences cellulolysis mechanistically?. Please support these claims better.
The nitrogen is reported to be linked with cellulose degradation by improving cellulase activity. We reflected this fact in the revised paragraph:
“Although our experiment wasn’t directly designed to solely test the effect of nitrogen on the effectiveness of decomposition, still, it was shown to be a considerable factor in determining the cellulolytic efficiency of BAGS as a bioproduct. Indeed, the gen2 BAGS, which was prepared with added nitrogen, had a significantly higher potential nitrification level than the gen1 BAGS. This suggests that BAGS activity is likely to be directly related not only to the cellulolytic component of the community, but to microorganisms associated with the nitrogen cycle. The nitrogen availability was shown to be linked with the cellulase activity during litter decay [89]. Therefore, we analyzed the presence of key nitrogen metabolism genes associated with nitrogen retention in MAGs [90,91]. ”
The authors claim that the Gracilibacteria–Chryseolinea is central to the results. However, there is no direct metabolic coupling analysis, and only presence/absence data are available. Please reformulate this claim.
We agree that we overestimated this claim. We merely were excited when we accidentally discovered this fact, so we thought of demonstrating it more visibly. However, we see that it is not central to the research and removed it from the abstract.
Figures 2A and 2B require zooming to interpret; please re-scale them.
Done
Figure 3 heatmap needs legend scaling
Done
Figures 4A and AB requires heavy explanation on why they are important to the results.
Removed from the body of the article in the supplement
Reviewer 5 Report
Comments and Suggestions for Authors
Abbreviations should not be used in the title.
Abstract:
Begin the abstract with the research objective, followed by the experimental design, presenting treatments and replications.
The methods presented should be concise.
Authors should focus on presenting the results.
Abbreviations must be defined before their use.
A recommendation should be included at the end of the abstract.
Keywords:
Do not repeat words from the paper title; Remove abbreviations.
Introduction
A hypothesis should be included at the end of the introduction.
Materials and Methods
Presenting this item after the results and discussion makes it somewhat difficult to understand what was evaluated. I believe this is standard practice for the journal.
Describe the experimental design more clearly: number of replicates per time and per generation; how the samples were synchronized between gen1 and gen2.
Specify quality parameters for amplicon and metagenome (filtering thresholds, rarefaction, Q-scores).
Briefly justify the choice of pipeline (dada2, SPIEC-EASI, WGCNA).
Describe how the obtained data were processed to obtain the results.
Results
Some figures require greater clarity (resolution, information in the legends, color scale, indication of statistical significance).
Author Response
Abbreviations should not be used in the title.
I know a lot of examples when abbreviations are used in the title, specifically when it concerns some product or tool (e.g.
Callahan, B.J.; McMurdie, P.J.; Rosen, M.J.; Han, A.W.; Johnson, A.J.A.; Holmes, S.P. DADA2: High-Resolution Sample Inference from Illumina Amplicon Data. Nat Methods 2016, 13, 581–583, doi:10.1038/nmeth.3869
Chaumeil, P.-A.; Mussig, A.J.; Hugenholtz, P.; Parks, D.H. GTDB-Tk v2: Memory Friendly Classification with the Genome Taxonomy Database. Bioinformatics 2022, 38, 5315–5316, doi:10.1093/bioinformatics/btac672.).
But if it is a general misconception, which should be improved, I agree to remove the preparation name from the title. Now the title sounds like “Search for the Active Bacterial Component of the Lignocellulose-Degrading Multispecies Biological Preparation”
Abstract:
Begin the abstract with the research objective, followed by the experimental design, presenting treatments and replications.
The methods presented should be concise.
Authors should focus on presenting the results.
Abbreviations must be defined before their use.
A recommendation should be included at the end of the abstract.
Thank you for the recommendations for the abstract improvement. It has been updated:
“The mobilization of complex microbial communities from natural resources can be a useful alternative to the use of single-species preparations when it comes to the decomposition of plant residues. However, the functioning and interaction of microorganisms within these communities remain largely unexplored. Our task was to investigate the cellulose-degrading community using the complex microbial preparation BAGS (biologically active “ground” with straw) as an example and define its active component. For this, monitored the succession of chemical parameters and taxonomic composition of the preparation during two consecutive rounds of its six-month composting process, varying in added nitrogen. The amount of added nitrogen significantly affected the performance of preparation, contributing to high cellulolytic activity, respiration, and ash content. The group of covarying ASVs (amplicon sequence variants), correlating with the mature phase of effective BAGS preparation, which developed by the fourth month of composting, was described. Metagenomic analysis of this phase revealed MAGs (metagenome-assembled genomes) corresponding to these ASVs, which contained genes for cellulose and aromatics degradation, as well as genes for nitrogen and sulfur pathways, including anaerobic nitrate reduction and thiosulfate oxidation. Thus, we propose that active bacterial component of BAGS, associated with the mature phase of effective preparation, occupied different trophic niches, not limited to cellulose degradation, which should be considered when designing natural or artificial microbial systems for the decomposition of plant residues.”
Keywords:
Do not repeat words from the paper title; Remove abbreviations.
Done
Introduction
A hypothesis should be included at the end of the introduction.
We modified the last paragraph of introduction and added hypothesis:
“The aims of the study were to define the mature phase of the BAGS preparation, reveal its active bacterial component, and describe its metabolic potential. To achieve this, we measured the performance of two BAGS generations using chemical and microbiological methods, performed sequencing of 16S rRNA gene amplicon libraries from different phases of BAGS incubation, and performed full metagenomic sequencing of the maturation phase of the preparation. The hypothesis was that changes in chemical parameters would help us identify the active phase of the preparation and its associated microflora. Metagenomic analysis, in turn, would reveal the diversity of the functional profile of this potentially active microflora. This knowledge would reveal the understanding of interaction within complex microbial cellulose-degrading communities.”
Materials and Methods
Presenting this item after the results and discussion makes it somewhat difficult to understand what was evaluated. I believe this is standard practice for the journal.
I totally agree, but it was a requirement for the other MDPI Journal. Luckily, Microorganisms support the idea that M&M should be before the results. So, the methods section was moved there.
Describe the experimental design more clearly: number of replicates per time and per generation; how the samples were synchronized between gen1 and gen2.
We had a single tub for each generation. Every month we collected one sample, but the analyses from this sample were performed in replicates: 2-3 for chemical analyses and 4 for the amplicon sequencing. The information was added to the materials section.
Specify quality parameters for amplicon and metagenome (filtering thresholds, rarefaction, Q-scores).
We have added additional information to the materials and methods section.
Briefly justify the choice of pipeline (dada2, SPIEC-EASI, WGCNA).
We use a modified pipeline based on dada2 to analyze amplicon data. Among denoisers, dada2 has the highest sensitivity, but it also makes a large number of errors. We believe that this is acceptable for use with soil microbiomes or similar substrates, such as our biopreparations.
We used two approaches to network analysis: WGCNA and SPIEC-EASI. WGCNA was chosen for its high degree of interpretability, ease of use, and validation capabilities during analysis. The analysis package is a convenient framework that we have previously used for time series analysis, in conjunction with normalization in the ANCOM-BC package. We used it to identify the active components of the community. SPIEC-EASI is a more complex, compositional approach to network construction. It was used for formal network description and identification of Gracillibacteria symbionts.
Describe how the obtained data were processed to obtain the results.
Results
Some figures require greater clarity (resolution, information in the legends, color scale, indication of statistical significance).
Resolution corrected, add color scaling on the heatmap, add replicates to the agrochemical plot.
Round 2
Reviewer 3 Report
Comments and Suggestions for Authors
The authors have addressed the requested modifications, and after a careful review, I am in favor of the manuscript’s publication.
Author Response
We would like to thank the reviewer for accepting our manuscript.
Reviewer 4 Report
Comments and Suggestions for Authors
authors made all the requested modifications
Author Response

(The authors gave the same response as above.)
